# Effect of pulse-current-based protocols on the lithium dendrite formation and evolution in all-solid-state batteries

V. Reisecker[1,2,11], F. Flatscher[2,3,11], L. Porz[3], C. Fincher[4], J. Todt[5], I. Hanghofer[6], V. Hennige[6], M. Linares-Moreau[7], P. Falcaro [7], S. Ganschow [8], S. Wenner[9], Y.-M. Chiang [4], J. Keckes [5], J. Fleig [10] & D. Rettenwander [1,2,3] ✉

Understanding the cause of lithium dendrites formation and propagation is essential for developing practical all-solid-state batteries. Li dendrites are associated with mechanical stress accumulation and can cause cell failure at current densities below the threshold suggested by industry research (i.e., >5 mA/cm$^2$). Here, we apply a MHz-pulse-current protocol to circumvent low-current cell failure for developing all-solid-state Li metal cells operating up to a current density of 6.5 mA/cm$^2$. Additionally, we propose a mechanistic analysis of the experimental results to prove that lithium activity near solid-state electrolyte defect tips is critical for reliable cell cycling. It is demonstrated that when lithium is geometrically constrained and local current plating rates exceed the exchange current density, the electrolyte region close to the defect releases the accumulated elastic energy favouring fracturing. As the build-up of this critical activity requires a certain period, applying current pulses of shorter duration can thus improve the cycling performance of all-solid-solid-state lithium batteries.

Providing future generations with sustainable and emission-free electrified transportation requires the development of energy storage concepts with specific energies of at least 400 Wh/kg[1]. This demand exceeds, however, the limit which can be reached by conventional Li-ion batteries (LiBs; ~250 Wh/kg)[2–5]. One viable solution to this issue could be the replacement of the flammable non-aqueous liquid electrolyte solution with an inorganic solid-state electrolyte (SSE), which, not only allows for an improved safety, but also the application of Li metal as the negative electrode, leading to a substantial increase in energy density (up to 50%)[3,5]. One of the most promising SSEs to enable such high performance solid-state Li batteries is cubic

$Li_7La_3Zr_2O_{12}$ (LLZO) and its variants. In comparison to other SSEs, it can provide a high room-temperature Li-ion conductivity approaching that of a conventional non-aqueous liquid electrolyte solution, whilst simultaneously showing good chemical and electrochemical stability toward Li metal and high-voltage cathodes[6–8].

However, to reach the promised performance, solid-state Li batteries must plate a large Li thickness of at least 15 μm or 3 mAh/cm$^2$ at a high (charging) current rate (>5 mA/cm$^2$) for a minimum of 1000 full cycles[5,9–12]. Today, under such conditions, these batteries invariably fail due to the formation of Li dendrites penetrating the SSE and causing a short-circuit[3,4,9,10,13]. One of the main reasons for this battery failure are

[1]Institute of Chemistry and Technology of Materials, Graz University of Technology, Graz, Austria. [2]Christian Doppler Laboratory for Solid-State Batteries, NTNU Norwegian University of Science and Technology, Trondheim, Norway. [3]Department of Material Science and Engineering, NTNU Norwegian University of Science and Technology, Trondheim, Norway. [4]Department of Materials Science and Engineering, Massachusetts Institute of Technology, Cambridge, MA, USA. [5]Department of Materials Physics, Montanuniversität Leoben and Erich Schmid Institute for Materials Science, Austrian Academy of Sciences, 8700 Leoben, Austria. [6]AVL List GmbH, Graz, Austria. [7]Institute of Physical and Theoretical Chemistry, Graz University of Technology, Graz, Austria. [8]Leibniz-Institut für Kristallzüchtung, Berlin, Germany. [9]Sintef Industry, Department of Materials and Nanotechnology, Trondheim, Norway. [10]Institute of Chemical Technologies and Analytics, TU Wien, Vienna, Austria. [11]These authors contributed equally: V. Reisecker, F. Flatscher. ✉e-mail: daniel.rettenwander@ntnu.no

current constrictions at the Li|SSE interface that arise from, e.g., (1) poor contact between the Li metal electrode and the SSE, (2) grain boundaries, or (3) void formation at the Li|SSE interface during Li metal stripping[9,10]. The resulting high local current densities lead to electro-chemo-mechanical stresses high enough to initiate Li penetration into the SSE, finally resulting in a short circuit and even cell failure[9,11,13].

To address this issue, various concepts have been explored like the introduction of Li-alloys/interphases (e.g., Sr, ZnO, and Mo)[14–16], additives (e.g., $Li_3PO_4$ or excess LiOH)[17,18] or interface engineering (e.g., introducing different atomic interlayers or creating 3D structures)[19,20], all of which are measures to enhance the SSE's wetting and/or increase the Li|SSE contact area. Applied on their own, these approaches are, however, not sufficient to meet the performance demand for practical applications. Only in combination with a constant heat supply, leading to a faster $Li^+$ diffusion[21], or the application of stack pressure to enhance the Li creep from the anode toward the interface[22], the necessary current density threshold becomes tangible. An increased operating temperature is, unfortunately, not applicable under all circumstances and additionally represents a constant energy drain. Furthermore, the application of stack pressure increases the risk of mechanical failure and is technologically challenging, as are some alloy formations[23]. To make use of the already achieved progress while bypassing critical temperature and pressure conditions, a different approach is needed that is compatible with the aforementioned methods. One such solution could be the alteration of the external current application. Pulsed currents are often utilized in electroplating of metals to achieve more uniform deposits[24–26] and have already been applied in $Na||O_2$ battery and $N_2O$ electrochemical reduction systems[27,28]. Current pulses therefore represent a potentially useful method for inhibiting Li dendrite formation[29–32].

Upon application of a direct current to a conventional LiB, Li-ions start to deposit throughout the whole electrode|electrolyte interface and can gather in certain areas causing concentration gradients. In case of a pulsed current program, where the current profile is interrupted by pausing/current-off times, the Li-ions have time to diffuse from regions of high concentrations toward regions of low concentrations (during said pausing times) which overall results in a denser and uniform Li-ion deposition[33–35]. Implementing a voltage pulse into the charging protocol of an all-solid-state high-voltage Li metal cell has already proven to reduce the interfacial impedance by refilling formed voids via Joule heating[36]. While a single pulse was not found to significantly enhance the current density limit, a repetitive application of pulses in a Li||LiFePO4 cell with conventional non-aqueous liquid electrolyte solution has turned out to produce dense microstructures. The application of conventional direct current on the other hand initiated rapid growth of porous Li film structures, degrading the cycling performance significantly[37]. Moreover, it has been shown that the mitigation of current constrictions by pulsed current waveforms enables a substantial suppression of the dendrite growth[34,38,39]. Another application is to intermittently apply a reverse current in a pulse form or a very small direct current (1–20 μA/cm²) to remove grown dendrites and receive a denser Li morphology[40,41]. Besides for morphological aspects, pulsed currents have also been studied as a measure to adjust the operating temperature of a battery by means of a self-heating mechanism[42,43].

Despite the potential of pulse plating to surmount the limitation imposed by Li dendrites in conventional LiBs, its effectiveness with respect to solid-state Li batteries has received only minor attention. Theoretical work has mostly focused on the effects of pulsed currents on non-aqueous liquid electrolyte solutions and the mechanisms studied therein are not necessarily transferable to solid electrolytes, where ions are spatially confined and much less mobile[29,34,35,38,39].

Specifically, it is not clear to which extent and how pulsed current waveforms can increase the so-called Critical Current Density (CCD) of solid-state batteries, which is the current density up to which safe cycling can be conducted without the formation of Li filaments[10,44,45] and what impact setting parameters like the pulse/pause ratio, frequency or applied current density have. Since the application of pulsed currents is an external electrochemical measure and does not interfere with any internal measures taken to increase the CCD, such as the application of alloys, additives, or interface engineering, it can be widely applied on top of such methods.

Herein, to analyze the impact of current pulses on different inorganic solid-state electrolyte morphologies, single crystalline $Li_6La_3ZrTaO_{12}$ (SC) and hot-pressed polycrystalline $Li_{6.4}La_3Zr_{1.4}Ta_{0.6}O_{12}$ (HP) cuboids of high geometrical and interfacial quality were prepared and subjected to different pulse plating protocols. A wide spectrum of techniques, such as focused ion beam, scanning electron microscopy, transmission electron microscopy, atomic force microscopy and electrochemical impedance spectroscopy were applied to retrieve a full sample profile. Afterwards, all samples were galvanostatically cycled in symmetrical Li cells using either direct or pulsed currents, whereby short circuits have been identified and tracked via operando Optical Microscopy.

This study reveals that MHz pulses enable up to a six-fold increase in CCD, leading to values as high as (6.6 - 0.1) mA/cm². The increase in

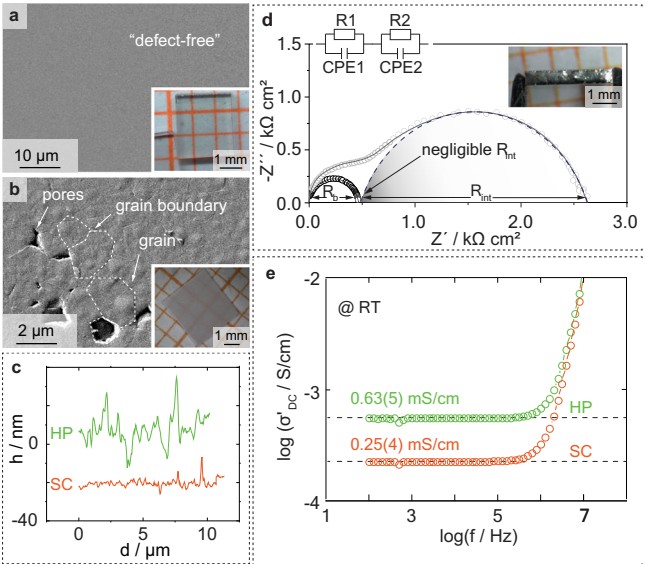

**Fig. 1 | Physicochemical and electrochemical characterizations of the inorganic solid-state electrolytes. a** (inset) Single crystalline $Li_6La_3ZrTaO_{12}$ (SC) specimen and **b** (inset) hot-pressed polycrystalline $Li_{6.4}La_3Zr_{1.4}Ta_{0.6}O_{12}$ (HP) specimen in top view after shaping and thorough polishing on millimeter paper. The samples were additionally investigated via scanning electron microscopy (**a, b**) prior to coating with Li:Sn. In the case of the SC sample no indication of any microstructural defect was found, whereas the HP pellet shows a very dense microstructure composed of grains with diameters around 2 μm and pores up to 1.5 μm. **c** Corresponding atomic force microscopy analyses were conducted to assess the defect concentration of the SC (orange) and HP (green) surface, and the measured height $h$ profile plotted against the distance $d$ covered. Note: for comparison reasons the topography profile of the SC sample was shifted by −20 nm. **d** Impedance data of an SC sample with (black) and without (gray) a proper surface treatment, measured in a symmetrical Li:Sn cell at 21 ± 1 °C without stack pressure. The inset demonstrates the homogeneity of the coating procedure after polishing. The negative imaginary part −$Z''$ is plotted against the real part of impedance $Z'$ in form of a Nyquist plot. In case of the polished sample, only the bulk contribution ($R_b$) can be properly identified, whereas the interfacial contribution ($R_{int}$) is negligibly small. Fits were calculated using the circuit shown on top, consisting of resistance elements R and constant phase elements CPE. Experimental data are represented by circular markers, fitted data by line markers. **e** Conductivity $\sigma$ is plotted against the frequency in form of a conductivity isotherm of an SC (orange) and HP sample (green), again, measured in a symmetrical Li cell at 21 ± 1 °C.

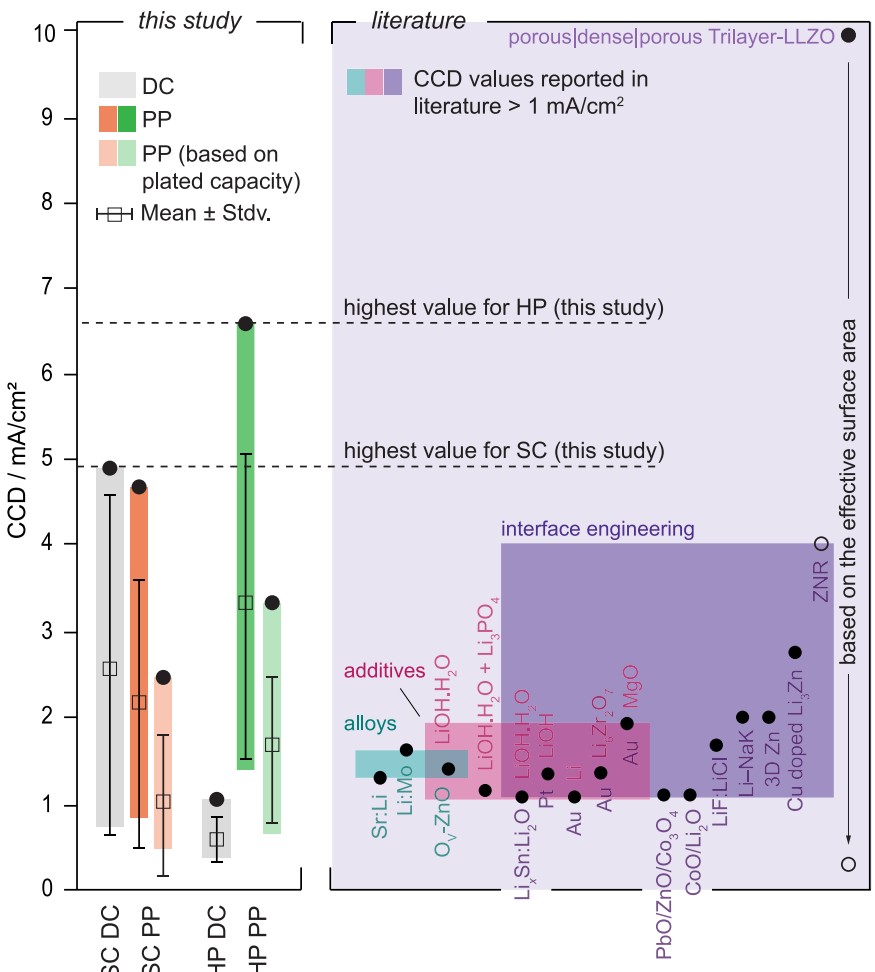

**Fig. 2 | Analysis and comparison of the experimental electrochemical measurements.** Ranking of the critical current density (CCD) values achieved for single crystalline (SC) and hot-pressed polycrystalline (HP) samples during either direct current application (DC; marked in gray) or pulse plating (PP; marked in orange and green for single crystalline and polycrystalline samples, respectively) operation. CCD values obtained during pulse plating are represented by black circular markers, whereas effective $CCD_{eff}$ values, considering the time increase during the measurement compared to the values when direct current has been applied, are plotted as white rectangular markers. The violet section shows CCD values above 1 mA/cm² reported in literature for different LLZO chemistries (excluding potentially soft shorted samples or values obtained at temperatures higher than room temperature and/or high stack pressure, to our best knowledge; based on refs. [14–20,23,55–63]. from left to right−see also Supplementary Table 5). Values were grouped into one or two of three categories depending on their strategy being either (1) the coating of an alloy onto the SSE (petrol), (2) application of an additive during the sintering procedure (pink) or (3) the engineering of the SSE interface by e.g., structuring or application of an additional interlayer (purple). For the rightmost value the full circle corresponds to the reported current density for the porous structure and the empty circle to the current density if the porous structure were planar.

CCD can be associated with the application of current pulses shorter than the time required to build up a critical Li activity near a defect tip located at the Li|SSE interface. Once reached, this critical Li activity leads to the structural destabilization and fracture of the SSE, which is accompanied by Li dendrite initiation and propagation.

## Results and discussion

### Ensure minimum sample-to-sample variation

One of the main challenges in studying the CCD in a reliable and reproducible manner is its dependency on microstructural features like grain boundaries, voids and flaws[46–48] or also chemical variations even within similarly prepared samples[49,50]. In order to reduce the impact of sample-specific parameters on the CCD measurements and guarantee comparable starting conditions for all experiments, special attention was paid to minimize sample-to-sample variations for each morphology type. Therefore, rectangular shaped SC and HP samples exhibiting similar sizes of roughly 3 mm × 3 mm × 0.5 mm (width, length, thickness) with densities of 100% and >99%, respectively, were prepared (see Fig. 1a, b, details regarding the sample preparation can

be found in "Methods" section and Supplementary Figs. 1 and 2). To achieve a minimal defect concentration at the electrolyte surface, a thorough surface treatment procedure consisting of multiple rotational and vibrational polishing steps was applied. Scanning electron microscopy images of the SCs did not reveal any macroscopic pores or larger scratches (Fig. 1a) present at the polished surface. The HP samples showed a dense microstructure with grains in the size of about 4 μm in diameter (Fig. 1b) and only minor pores up to about 1 μm in diameter. In both cases, a smooth surface profile was achieved with root mean square (RMS) roughness values of (3 ± 1) nm and (8 ± 3) nm for the SC and HP samples, respectively (Fig. 1c and Supplementary Fig. 3).

Besides its morphological dependency, the CCD has been found to be heavily influenced by current constrictions along the Li|SSE interface[9,10]. Current constrictions are related to missing contact associated with insufficient Li wetting or surface contaminants for instance.

Moreover, the ratio between the conductivity of the SSE and the exchange current density has been shown to have a significant impact

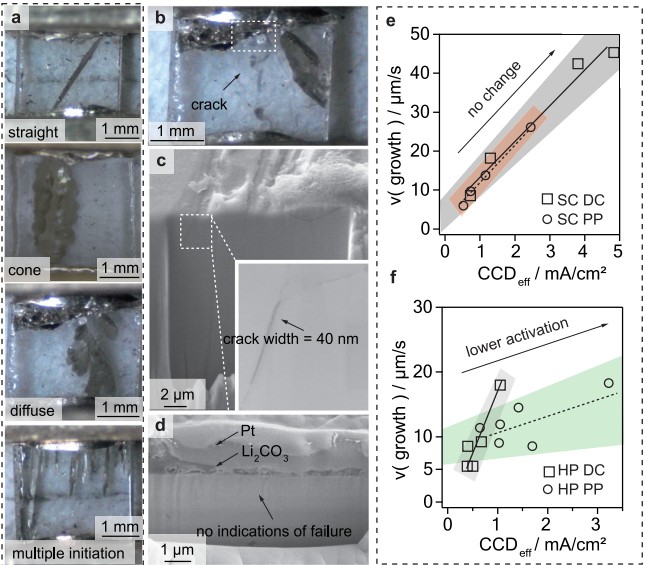

**Fig. 3 | Microscopy investigations of the Li metal depositions. a** Overview of different Li dendrites observed during optical tracking of single crystalline samples being of either straight, cone or diffuse, type. In some cases, multiple dendrite initiation was spotted, where dendrites started growing from several initiation sites at the same time. The dendrite growth can be seen in motion in Supplementary Videos 1–5. Crack analysis performed on an SC (**b–d**) and HP (**f**), cut parallel to the formerly coated electrolyte surface with a focused ion beam. For the single crystalline sample small subsurface cracks at a width of 40 nm were found, whereas for the hot-pressed no cracks large enough to resolve were found. Growth rates plotted against the effective critical current density (CCD$_{eff}$) values in case of direct current (DC) or pulsed current (PP) operation for single crystalline LLZTO (SC, **e**) and hot-pressed polycrystalline LLZTO (HP, **f**). The trends have been colored in gray (DC), orange (SC PP) and green (HP DC) for visualization purposes.

on the uniformity of metal deposits[51]. The area specific resistance (ASR) directly reflects both, the exchange current density (according to Butler-Volmer) and the degree of physical contact[9,45]. Hence, a reliable experimental platform has to ensure similar ASR values, as well as similar Li-ion conductivities within one experiment series. Therefore, potential contaminants at the SSE surface, such as LiOH and Li$_2$CO$_3$, have been removed by a proper polishing sequence with a subsequent heat treatment at 400 °C in Ar atmosphere[52]. Moreover, a molten Li:Sn electrode (70:30 wt%) has been used to improve the wetting of Li on the SSE[53]. The combination of both approaches led to a reproducible reduction of the ASR value down to negligible values (see Fig. 1d and Supplementary Note 1). This is elaborated further in Supplementary Tables 1 and 2 as well as Supplementary Fig. 4. The variation in Li-ion conductivity across the SC and HP samples was investigated by plotting the electrochemical impedance spectroscopy data in a Bode-like fashion (Fig. 1e). Both conductivity isotherms reveal a single direct current plateau, which, in the case of the SC, can be related to the bulk conductivity due to the lack of grain boundaries. In case of the HP sample, the grain boundaries appear to not significantly contribute to the total resistance of the SSE, resulting in a single contribution as well. The averaged conductivities and deviations of the SC and HP samples amount to $(0.25 \pm 0.04)$ mS/cm (mean $\pm$ std) and $(0.63 \pm 0.05)$ mS/cm at $(21 \pm 1)$ °C, respectively, pointing toward minor sample-to-sample variations in terms of resistance. Additional information can be found in Supplementary Figs. 5 and 6.

In addition to these parameters, experimental conditions, like temperature[21] and pressure[54] impact the CCD. Increasing either of the two can significantly elevate the CCD value, e.g., by enhancing the creep of Li metal and its diffusivity. Therefore, all experiments were conducted at a constant temperature $(21 \pm 1)$ °C and without the

application of significant stack pressure (less than 3 kPa, see Supplementary Note 2).

## The critical current density

Different pulsing sequences were tested in preliminary experiments with polycrystalline LLZTO (PC) pellets prepared by conventional solid-state synthesis, where the pulse-pause ratio was varied from 1:1 to 1:10 in the ms-range. Whereas these cycling conditions did not improve the electrochemical performance of the PC pellets, reducing the respective timeframes to the μs-regime enhanced the CCD values achieved in contrast to direct current operation for a 1:1 pulsed current condition (see Supplementary Note 3 and Supplementary Table 2). The same pulsing sequence was then applied to the SC and HP samples and compared to the CCD values achieved under direct current operation (see Fig. 2, Supplementary Note 4 and 5 and Supplementary Tables 3 and 4). Since the pausing time in the pulsing sequence prolongs the overall cycling time, an "effective" CCD value (CCD$_{eff}$) was defined on the basis of the plated capacity for better comparison. For a 1:1 pulsing sequence, taking twice as long as direct current cycling, the obtained CCD value was therefore divided by two. Prior to any cycling experiment conducted with pulsed waveforms, the efficiency was evaluated by galvanostatic titration and plating experiments. The measurements revealed that pulsed currents down to the μs-range can achieve efficiencies up to 100% (see Supplementary Note 6 and Supplementary Figs. 7 and 8). In the case of the SC and HP samples, the cycling profile was complemented by simultaneously taken optical microscopy images, easing the identification of Li dendrite formation (details are given in Supplementary Note 4, Supplementary Figs. 9 and 10 and Supplementary Videos 1–5).

For the SC samples, CCD values up to $(4.95–0.05)$ mA/cm$^2$ (current at potential drop–current step size, taken as range containing actual CCD value) have been achieved by using direct current application. Compared to similar samples lacking an extensive surface treatment[45], this represents an increase of around 170%, thereby highlighting the sensitivity of the CCD on the electrolyte's surface condition. In Fig. 2 and in Supplementary Table 5 previously reported CCD values are summarized (note: based on the effective surface area; values related to potential soft shorts, higher temperature and high stack pressure are excluded). Values up to 4 mA/cm$^2$ have so far been achieved by using alloys (e.g., Li:Mo, Sr:Li), additives (e.g., LiOH·H$_2$O) and/or interface engineering (e.g., MgO, Cu-doped Li$_3$Zn)[14–20,23,55–63]. While the SC samples exceed these values by 1 mA/cm$^2$, they are, however, not practicable for large-scale production, due to possible cost and manufacturing concerns. Applying a direct current protocol to the HP samples, however, led to average CCD values below 1 mA/cm$^2$, proving that the presence of grain boundaries and small pores reduces the CCD (see Supplementary Tables 3 and 4). When switching to pulsed currents, no significant difference in electrochemical performance could be found for the SC samples. While the maximum CCD reached a similarly high value of $(4.8–0.1)$ mA/cm$^2$, the overall average was lower, taking on a value of $(2.1 \pm 1.6)$ mA/cm$^2$, as opposed to $(2.7 \pm 2.0)$ mA/cm$^2$ for direct current cycling. Based on the CCD$_{eff}$ of $(1.0 \pm 0.8)$ mA/cm$^2$ it becomes evident that in the case of SCs, pulsed currents cannot effectively enhance the CCD. For HP samples on the other hand, an improvement could be observed with a maximum CCD of $(6.6–0.1)$ mA/cm$^2$ and an average of $(3.3 \pm 1.8)$ mA/cm$^2$. Even based on the CCD$_{eff}$ of $(1.7 \pm 0.9)$ mA/cm$^2$, pulsed currents can by far outperform cells cycled under direct current conditions reaching average CCDs around $(0.6 \pm 0.3)$ mA/cm$^2$.

These results indicate that MHz pulsed currents can mitigate critical defects up to a certain current density range, enabling current densities as high as $(6.6–0.1)$ mA/cm$^2$, and an increase by a factor of three concerning the plated capacity. Notably, despite our efforts to mitigate as many influencing factors as possible, large variations in

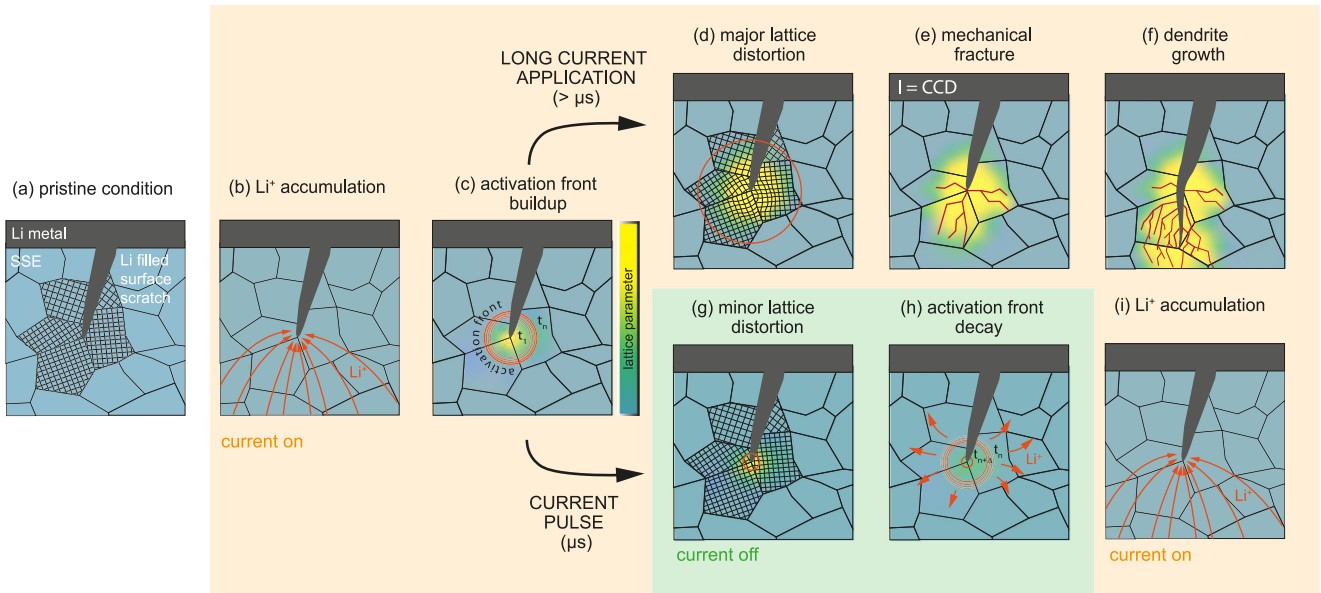

**Fig. 4 | Proposed mechanisms in constant current and pulsed current electrochemical experiments.** Flow diagram showing the proposed mechanistic differences between direct current and pulsed current (µs) operation of a solid-state cell employing a Li metal anode: **a** pristine condition of the Li|SSE interface exhibiting a surface scratch of the SSE filled with Li, along with the lattice orientations of the adjacent grains near the defect tip. **b** Once the current is switched on Li-ions start to concentrate near the defect tip and **c** cause the buildup of an activation front, increasing over time $t_n$. LLZTO is thereby locally reduced and the lattice parameter in this region changed. **d** In case of direct or long current pulse (>µs) application this continuous lattice distortion causes a continuously increasing amount of pressure which, at some point, is released in form of (**e**) mechanical fracture. As a consequence, Li is plated along the new cracks (**f**) and drives the mechanical fracture even further until a short circuit is caused. **g** In case of short current pulses (µs) the time for the activation front buildup is short enough to just cause a minor lattice distortion before the current is switched off again. **h** The accumulated Li-ions start to diffuse and distribute into the neighboring regions, hindering a significant pressure to arise. **i** Afterwards, the current is switched on again like in (**b**) and the process repeats itself.

CCD values were still observed. This suggests that studies on CCDs should, in general, be statistically evaluated to ensure reproducibility.

## Li dendrite characteristics

During optical tracking of Li dendrite initiation and propagation in SCs, a multitude of different dendrites has been observed, which were essentially grouped into one of three different categories similar to the classification used by Kazyak et al.[64]: straight, cone or diffuse type (see Fig. 3a). The straight type is characterized by a 2D Li deposition throughout the SSE, whereas the cone type branches out at the sides causing more of a 3D growth pattern. In the case of the diffuse type, Li did plate in a flowing manner followed by a macroscopic fracture of the electrolyte. For the multiple initiation type, filament growth started at several initiation sites at once and continued growing at the same time. Figure 3b–d shows the analysis of the crack widths of a typical shorted SC and HP sample using a focused ion beam-scanning electron microscope. In the SC sample sub-surface cracks with widths of about 40 nm were observed, whereas the HP sample did not show any signs of intra- and/or intergranular crack propagation. This observation could, however, also be related to the formation of dendrites below the resolution limit of the scanning electron microscope. As all dendrite types were observed in both cycling programs, no clear correlation between the current waveform and the respective shape of the dendrite could be found. In most cases, either a straight or cone-like Li propagation was observed, whereas the diffuse type was encountered very rarely. In contrast to the observations Kazyak et al.[64] have made, the propagation of each dendrite type was accompanied by a decrease in potential. Overall, initiation sites were found to be randomly distributed across the whole interface and only numerous in case of high CCDs (Fig. 3a bottom). As for the HP samples, Li dendrite formation was not observed to follow a single initiation event, but multiple ones, starting simultaneously throughout larger areas of the Li|SSE interface (Supplementary Fig. 11). Next to so-called hard shorts, soft shorts, in some cases exhibiting seemingly reversible Li deposition, have also been observed but were not further studied herein (see Supplementary Note 7 and Supplementary Figs. 12 and 13).

## The growth kinetics

Growth rates were determined by dividing the distance between the electrodes overcome by the time needed to reach the opposing electrode. Figure 3e, f shows the average growth rates of both morphology types obtained under either direct or pulsed current conditions. A linear relationship between the CCD and the growth rate becomes evident, which is in agreement with previous studies[64]. In the case of the SCs, Li deposition rates as a function of CCD progressed at speeds of 8.4 µm/(s(mA/cm²)) and 10.5 µm/(s(mA/cm²)) under direct and pulsed current operation, respectively, indicating little dependence of the Li deposition mechanism on the pulse waveform. Applying the same approach to the HPs, pulsed currents were observed to reduce the growth rate increase with increasing CCD from 13.7 µm/(s(mA/cm²)) to 1.1 µm/(s(mA/cm²)). These observations suggest that Li propagation follows a different mechanism in polycrystalline samples under pulsed currents as opposed to in single crystalline ones.

The growth rate of a dendrite can be expressed by $dx/dt$ and based on our observation takes on a value of about 10 µm/s at 1 mA/cm². The current density at the tip, required to enable growth rates up to 10 µm/s, can be derived from Eq. (1):

$$j = \frac{I}{A} = \frac{\left(\frac{dV}{dt} \times \frac{1}{V_m^{Li}}\right) \times F}{A} = \frac{\left(A \times \frac{dx}{dt} \times \frac{1}{V_m^{Li}}\right) \times F}{A} = 7\,\frac{A}{cm^2} \tag{1}$$

The total current flowing is denoted by $I$, the molar volume by $V_m^{Li}$, which amounts to 13.148 cm³/mol and $F$ represents the Faraday constant. Note that the tip area of the dendrite $A$ cancels out in the equation, which means that the current density is independent of the

size of the dendrite in a first approximation. If the current for the entire cell is much larger than the current needed for driving the dendrite, which is typically the case evidenced by no or only minor voltage drops before a short circuit, such an approximation can be made. According to this relation, the current densities needed to achieve the observed growth rates are in the range of 7 A/cm$^2$ when considering the required charge transfer to provide the necessary volume of Li. This value is larger than the exchange current densities reported for LLZO so far, which lie around 0.3 A/cm$^2$ [65]. Hence, the high current density causes a significant polarization at the dendrite tip during plating with detrimental consequences.

### The mechanism

As previously mentioned[46,66], the propagation of dendrites through ceramic electrolytes is suggested to be associated with the geometrical constraint for Li within a defect. Due to this constraint, a critical pressure builds up as soon as the defect is filled, causing a protrusion of Li into the SSE. This failure occurs instantaneously with the applied current (see finite element calculation described in Supplementary Note 8 and Supplementary Figs. 14 and 15), which would indicate that the longer pulses in the µs-range should not have any implications on the propagation of Li dendrites. The experiments with the HP samples, however, showed that the CCD is increased by a factor of six, whereas the Li growth rate decreases by one order of magnitude when switching from direct to MHz pulsed currents. Hence, the previously proposed mechanism of mechanical failure associated with the penetration of Li needs to be revisited to describe our observations.

Our main hypothesis is that the buildup of a Li-ion activity front in the neighboring region of a defect tip is responsible for a critical pressure buildup, eventually followed by fracture of the SSE and dendrite propagation. As the buildup of this critical Li-ion activity does not occur instantaneously once the current is switched on, current pulses of short enough duration can prevent the incident of this case. In this sense mechanical fracture actively drives the formation of Li dendrites rather than merely accompanying it. This suggested new mechanism not only explains the formation of Li dendrites under DC operation (see Fig. 4a–f) but would also be in accordance with the electrochemical performance increase observed for pulsed current operation (Fig. 4a-c, g–i), as elucidated in the following.

In its pristine condition (Fig. 4a), a solid-state cell employing a Li-metal electrode and polycrystalline LLZTO electrolyte will show surface defects like scratches (as observed in Supplementary Fig. 3) ultimately filled with Li after construction. In order to extend cracks within an SSE and allow the propagation of Li dendrites, bonds need to be broken. We suggest that the corresponding driving force is introduced locally and not as approximated previously by a homogeneous load[46]. The required energy gain to overcompensate the energy needed to break bonds can come from (1) the release of the chemical driving force from a Li chemical potential (or activity) in the SSE being above that of Li metal, and (2) the release of elastic energy due to an enhanced Li activity near the dendrite (and crack) tip. Both energy contributions emerge from the same phenomenon, being the accumulation of Li-ions and electrons in close proximity to the Li-filled crack tip once an electrode overpotential is applied (Fig. 4b). This is in accordance with Han et al. where a substantial enhancement of the electronic conductivity is found close to the Li electrode upon polarization[67]. In Solid-State Ionics this is well known as stoichiometry polarization or Wagner-Hebb polarization, where a blocking of ionic charge carriers at the electrode is present[67,68]. This is valid, as long as the critical conditions leading to the initiation and growth of Li dendrites are not met. Then, LLZTO is locally reduced and thus the electron concentration enhanced, driven by the very high current densities of beyond 1 A/cm² (see Eq. (1) above).

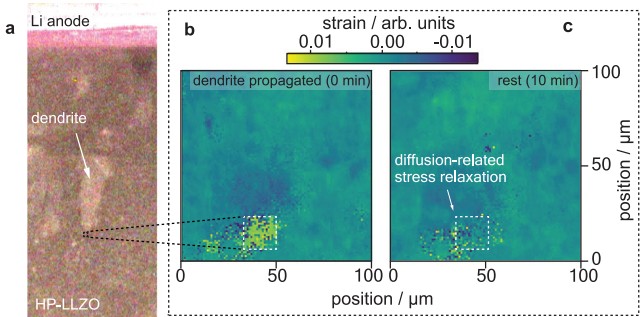

**Fig. 5 | Operando microscopy and X-ray measurements of the Li metal and solid-state electrolyte. a** The dendrite is highlighted in an optical microscope overview image, while the mapped region of 10 × 10 µm is marked as a black-dashed square. The corresponding operando synchrotron X-ray nano diffraction strain maps show the region ahead of a Li dendrite directly after its growth and after a waiting time of ~10 min. A single grain is highlighted and exhibits a deviatoric strain of ~0.0045 (**b**, 0 min), and −0.0015 (**c**, 10 min), equivalent to a change in deviatoric stress of ~750 MPa. This relaxation was not detected when comparing the state during the dendrite's growth and to the state directly thereafter. The grain is located ~8–10 µm in front of the assumed dendrite tip.

The local stoichiometry change (i.e., additional Li-ions and changed ionic valences due to additional electrons, Fig. 4c) induces a change of the local lattice parameter similar to, e.g., doped ceria (i.e., SSE for solid oxide fuel cells) under very reducing conditions[69]. This change only arises near the defect-tip region and is therefore suppressed by the mechanical constraints toward the bulk part of LLZTO, which is unaffected by Li activity changes (Fig. 4d). Accordingly, a substantial elastic energy builds up until a critical Li activity is reached resulting in stress relaxation by fracture (Fig. 4e) followed by Li dendrite propagation (Fig. 4f). Further evidence for this theory is given by observations of grain relaxation in an HP sample (Fig. 5a) measured by operando synchrotron XRD (1) directly during dendrite growth (Fig. 5b) and (2) after 10 min of current pause (Fig. 5c), allowing for diffusion. These measurements highlight the apparently strong mechanical sensitivity of LLZTO toward the local Li activity. Specifically, the 2nd-order (grain-average) deviatoric strain was observed to change from ~0.0045 to ~ −0.0015 within a single grain located between 8 to 10 µm away from the dendrite tip within 10 min. This corresponds to a difference in deviatoric stress on the order of 750 MPa. Other phenomena such as crack extension would act much faster, i.e., with the propagation speed of phonons, and can therefore be ruled out as explanation for this observed relaxation. Similar behavior can be also found, e.g., in $LiCoO_2$[70], and Si[71] where the insertion of Li-ions into the particle causes a volume mismatch between the new phase near the surface and the existing phase in the bulk once the relaxation kinetics is slower than the transfer rate. This volume mismatch causes a high enough chemo-mechanical strain to induce plastic deformation, mechanical fracturing and even amorphization. This effect can also be found in, e.g., fracture of rock formations as a consequence of water uptake, which could be seen as an analogous mechanism. The change of the unit cell upon incorporation leads to an expansion or shrinkage, inducing stresses which eventually cause the stone to crack[72].

Overall, these arguments strengthen the theory that dendrite propagation is promoted by the mechanical weakening of the SSE (i.e., decrease of the fracture resistance) at the crack tip caused by the constant increase in Li activity under current application. The time dependence of this process can be described by the ambipolar (chemical) diffusion coefficient of Li-ions in LLZTO. The exact time and space thereby depend on the geometry of the defect and the location at which Li-ions are injected and where electrons originate from[73]. Contrary to the concept of Sand's time, which has been used to

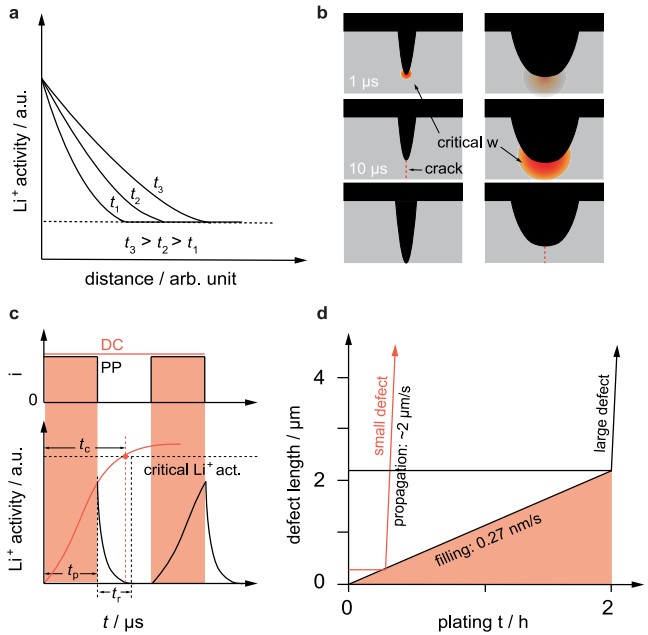

**Fig. 6 | Trends and observations concerning Li activity front and defect size.**
**a** Activity front at SSE|Li interface and growth of rim with increasing time (arbitrary chosen times $t_1$, $t_2$ and $t_3$ where $t_1 < t_2 < t_3$). **b** Schematics illustrating that a smaller activity front (red) with a critical width $w$ is sufficient to crack up the environment of small defects, (black) in the solid electrolyte (gray) whereas a thicker activity front is needed for the same result in case of bigger defects. It therefore takes for a longer time to induce fracture in the vicinity of big defects as opposed to smaller ones. **c** Increase of Li activity as a function of plating time between pulse plating (PP) and direct current (DC) application. The illustration shows that if the current is applied for a shorter amount of time ($t_p$) than what is needed to reach the critical Li activity ($t_c$), a structural degradation can be avoided/delayed, while relaxing for $t_r$. **d** Time required until a defect gets critical (i.e., filled completely) by considering a plating rate of 0.27 nm/s (assuming a current density of 0.2 mA/cm²), demonstrating that smaller defects are most critical in terms of filament initiation and propagation. Further elaborated in Supplementary Note 9.

describe the transition from mossy Li growth to dendritic growth in liquid cells via the formation of a Li-ion depletion layer and subsequent Li-ion deposition in current hotspots[74,75], we postulate a local increase of Li-ions. As already mentioned, this increase in Li-ion activity takes place at a defect near the Li|LLZTO interface (Fig. 4b), whereby in a first approximation a few assumptions can be made. On the one hand that there is an equilibration of electrons and on the other hand that any change of the interfacial double layer and any additional contribution of the interfacial charge transfer to the kinetics (i.e., considering diffusion rather than the Butler-Volmer limited case, due to the local constrain of Li metal in a filled crack tip) can be neglected. Hence, the applied overpotential directly transfers to a locally enhanced Li activity creating an activity front at the Li dendrite tip. The diffusion into the surrounding LLZTO regions (Figs. 4b and 6a), is determined by the chemical diffusion coefficient of Li-ions, $D$. Assuming radial symmetry for diffusion with the concentration C(R) at the center, a constant $D$ value and concentrations instead of activities, the additional Li-ion concentration $C(r,t)$ at radius $r$ and time $t$ in the corresponding region can be determined with Eq. (2):

$$C(r,t) = C(R) \cdot \mathrm{erf}\sqrt{r/Dt} \qquad (2)$$

The quantitative relation between the additional Li-ion concentration and the activity of Li (or the overpotential) is not known. However, the effective thickness $w$ of such a diffusion front with the

diffusion coefficient $D$ and the time $t$ can be approximated according to Eq. (3):

$$w = \sqrt{Dt} \qquad (3)$$

It shows that this "rim", having an enhanced Li-ion activity, increases with time (Fig. 6a). From a mechanistic point of view, the tip radius $R$ defines the stress concentration at a flaw. We suppose that to reach the critical condition for dendrite growth, a certain Li-ion activity is needed at the tip. For example, based on Eq. (3), it will take 100 times longer to reach critical conditions for a ten times larger $R$ (Fig. 6b). Therefore, a tip with a large radius $R$ (e.g., pores, scratches) requires a thicker activity front as opposed to a sharp tip (e.g., grain boundaries). In other words, the time required to reach the critical activity front is shorter for a tip with smaller $R$. Since the accumulation of a critical activity front at a dendrite takes place in the sub-ms range (depending on the applied current), this time dependence does not play a significant role for direct current charging.

In the case of pulsed currents, however, this time dependence becomes significant. For 1 MHz pulses the activity front can build up for just one microsecond, which is too short to reach critical conditions and only a minor lattice distortion takes place (see Figs. 6c and 4g). In the subsequent microsecond the Li-ion activity front can then relax again during the pausing time (Fig. 4h). As the current is switched on again, a similar starting condition is given and the cycle repeats itself (Fig. 4i). Hence, the propagation of Li dendrites is suppressed by taking away the driving force, i.e., fracture of the SSE at high Li-ion activity.

This explains the increase in CCDs and lower propagation rates of Li dendrites in case of the HP samples when pulsed currents are applied and why the SC samples remain unaffected under the same conditions. Considering that grain boundaries are sharp defects with lower fracture toughness compared to the bulk, Li dendrites are predominantly formed in these regions (see detailed discussion on the failure likeliness of different types of defects and defect sizes in Supplementary Note 9 and Supplementary Fig. 16). Hence, when the pulsed current frequency is sufficiently high, Li penetration will then take place, e.g., along grain boundaries at much higher currents (or any other defect for which the time required to reach a critical Li activity front is less than the pulse frequency) explaining the six-fold increased CCD for HP samples. The lower propagation rate during pulsed current application can be associated with local fluctuations in Li-ion transport and Li deposition rates[67]. These cause stress accumulation at the branch tips, which are created once Li starts to penetrate the SSE. Considering that the branch tips are "sharper", as opposed to the primary tip, they are most sensitive to failure (see Fig. 6d). Consequently, dendrites under pulsed conditions must pass a longer pathway, hence, require more time until a short-circuit occurs. Due to the absence of grain boundaries, SCs naturally short at higher current density values approaching those of HP under pulsed current conditions. At these current densities, however, 1 MHz pulses are not sufficient anymore to mitigate the build-up of a critical activation front, keeping the CCD unaffected by pulsing. Hence, the application of pulsed currents appears to be particularly powerful for improving the behavior of industrially relevant polycrystalline solid-state electrolytes.

To this date, lithium dendrites remain the weak point of solid-state Li batteries and hinder their implementation in practical electrochemical energy storage[2,3]. In order to overcome the critical current density (CCD) barriers set by industrial researchers (>5 mA/cm²) and become a competitive option for electric vehicles, different strategies are needed to exceed this limitation. Herein, we demonstrate that the application of 1 MHz-pulsed currents increases the CCD by a factor of six, leading to values as high as 6.5 mA/cm², thereby exceeding values reported in literature so far. To understand the origin of this

improvement, the preexisting mechanism of Li dendrite formation must be extended. We propose that an enhanced Li-ion activity close to the filament tip arises once Li deposition within a defect is limited by geometrical constraints, which, in turn, causes the effective current density at the crack tip to exceed the exchange current density. The increase in Li activity is accompanied by a lattice expansion that is constrained toward the bulk causing a buildup of elastic energy. Once a critical current has been reached, this energy is released by fracture of the ceramics. Since the buildup of a critical Li-ion activity requires a certain time, the application of current pulses with shorter durations can be used to extend the stability range of the solid-state electrolyte, and therefore increase the achievable CCD. We speculate that a combination of pulsed current waveforms in combination with other established methodologies, like the application of interlayers or increased interfacial surface areas, can significantly boost the performance of solid-state Li batteries.

## Methods

### Sample preparation

For this study, three different model systems of LLZTO were used. For preliminary tests, e.g., to determine coating and plating behavior and find suitable pulsing parameters, polycrystalline $Li_{6.5}La_3Zr_{1.5}Ta_{0.5}O_{12}$-pellets (PC) were prepared via solid-state synthesis. Therefore, stoichiometric amounts of $La_2O_3$ (Alfa Aesar, CAS No.1 preheated for 8 h at 900 °C), $ZrO_2$ (Millipore Sigma, CAS No. 1314-23-4) and $Ta_2O_5$ (Alfa Aesar, CAS No.1314-61-0) were mixed with LiOH (Alfa Aesar, CAS No.1310-65-2) in 10 wt% excess to account for any Li losses during synthesis. The powder was then wet milled with isopropyl alcohol (Sigma-Aldrich, CAS No.67-63-0; 15 ml) in 100 ml Zr jars with Zr balls for 6 h at 400 rpm. The finished slurry was dried in a Nabertherm oven at 60 °C followed by a calcination step (950 °C, 6 h) in an Al crucible. The sample was again wet milled with isopropyl alcohol (6 h, 400 rpm) to obtain a fine powder and isostatically pressed into green bodies of 10 mm diameter by applying a load of 5 tonnes for 1 min. The pellets were transferred to a Pt crucible, stacked and each covered in calcined power to avoid Li-evaporation. The crucible was again transferred to the oven and subjected to a two-step sintering procedure (950 °C for 0.5 h, 1180 °C for 16 h). A more detailed description of this procedure can be found in ref. 76. For the main cycling experiments hot-pressed, and therefore denser, polycrystalline $Li_{6.4}La_3Zr_{1.4}Ta_{0.6}O_{12}$ (HP) pellets were purchased from Toshima Manufacturing Co., Ltd. Materials System Division. Finally, single crystalline $Li_6La_3Zr_1Ta_1O_{12}$ pellets (SC) were prepared via the Czochralski pulling technique. Composition of the starting melt was stoichiometric with an additional excess of $Li_2O$ of 20 mol%. The raw materials, $Li_2CO_3$ (99.99% Merck), $La_2O_3$ (99.999% Fox Chemicals), $ZrO_2$ (Puratronic© Johnson Matthey), and $Ta_2O_5$ (99.999% Alfa Aesar) were weighed, mixed, isostatically pressed at 2 kbar, sintered for 6 h at 850 °C, ground, pressed again, and sintered for 6 h at 1230 °C. For the growth process, this starting material was melted in an inductively heated, 40 ml iridium crucible enclosed by alumina ceramic insulation in a pure $N_2$ ambient environment. After melt homogenization a thick iridium wire was immersed in the melt to initiate crystallization. With some material attached, the wire was slowly pulled upwards (0.5 mm h$^{-1}$) and the generator controlled by the automatic diameter control routine of the pulling station. After the growth was completed, the crystal was withdrawn from the melt and cooled down to room temperature in 15 h. The obtained crystal had a length of 40 mm at a diameter of 15 mm. The single crystals were compressed along a [150] direction which allows a maximum Schmid factor of 0.38 for the <111> {1−10} slip system. The influence of crystal orientation was excluded by cross-check experiments with a compression axis tilted by 45°. Testing in these two directions makes sure that any slip system is oriented with a Schmid factor >0 in at least one experiment.

The polycrystalline pellets were used for proof-of principle and coating experiments, so no special attention was paid to keep the surface roughness and defects to a minimum. Therefore, the cylindrical pellets were simply sanded and polished using up to 4000# grit SiC paper and a polishing cloth (Supplementary Figs. 2a and 4a). The SC and HP samples on the other hand were cut into geometrical cuboids with a diamond saw to obtain dimensions of ~2.7 mm × 3.3 mm × 0.5 mm and 2.3 mm × 2.5 mm × 0.8 mm (width, length, thickness), respectively. Special attention was paid to assure a parallel arrangement of both interface areas (Supplementary Fig. 2b) to avoid stress concentration at elevated regions when clamped into the measuring setup. Afterwards, the samples were glued (with Crystalbond) to a steel ingot for mechanical stabilization and polished in a multi-step mechanical and vibrational polishing procedure. To get rid of any surface contaminants, all samples were finally heat-treated at 450 °C for 3 h (heating rate 5 °C/min) in Argon atmosphere according to a study executed by Sharafi et al.[52] and wiped down with the polishing cloth directly prior to coating.

### Coating

In order to achieve good contact between the metal anode and the SSE throughout the whole interface, various Li:M-alloys (M = Sn, Zn, Na) were tested instead of pure Li to improve the wetting behavior and therefore reduce the interfacial resistance. In addition, combinations with Carbon (C-) and Au-interlayers were investigated to check for further improvements (Supplementary Fig. 4c and Supplementary Table 1). All tests were carried out with the PC samples. Interlayers were applied after the final polishing step in an Ar-filled glovebox from Braun ($O_2$ and $H_2O$ levels below <1 ppm) at (21 ± 1) °C directly prior to application of the Li:M alloy. The C-layer was applied by abrading a pure graphite crucible onto the pellet, whereas the 10 nm-thick Au-layer was sputter-coated in a Leica EM QSG100. The alloys were heated to around 250 °C and the pellet immersed for up to 10 min. Residual Li:M alloy at the sides of the pellet were removed by sanding, such that only the top and bottom sides were coated (see Supplementary Fig. 4b, d, e). While interfacial resistances were comparably low for all alloys, the Li:Sn-alloy stood out the most in terms of success rate and ease of handling, which is why this coating approach was chosen for all later measurements. In case of the SC and HP samples, only the polished sides (Supplementary Fig. 2b, green arrows) were dipped into the Li:Sn bath for around one second.

### Physicochemical characterization

For the synthesized polycrystalline and purchased hot-pressed polycrystalline samples, phase purity was checked via powder X-ray diffraction. The measurements were carried out with a Rigaku MiniFlex X-ray diffractometer with a step size of 0.1°/min in a scanning window ranging from 10 to 60° (see Supplementary Fig. 1).

Focused ion beam cross-sectioning and scanning electron microscopy imaging of cracks after failure were carried out using a FEI Helios G4 dual-beam instrument. The samples were placed in a sealed container with Argon atmosphere in the glovebox, transferred to the electron microscope and installed as quickly as possible with an estimated air exposure of 1 min. Additionally, atomic force microscopy measurements were conducted on polished SC and HP samples to analyze their surface roughness. Topography images were acquired with a Cypher ES Atomic Force Microscope from Oxford Instruments (Asylum Research), equipped with a Si cantilever from Budgetsensors (model Tap300GD-G, resonant frequency 300 kHz) and operated in tapping mode under inert gas (Ar).

Operando synchrotron X-ray nanodiffraction studies were carried out at the nanofocus extension of beamline ID13 at the European Synchrotron (ESRF)[77]. Images of the experimental setup can be found in Supplementary Fig. 17. HP samples were placed on microscope

slides and mapped by an X-ray beam of 0.09537 nm wavelength ~80 nm in diameter. The region of interest was chosen to include an area of $10 \times 10\,\mu m$ in size placed directly ahead of a growing Li dendrite, identified using an optical microscope. Diffraction patterns from the gauge volume ~80 nm × 80 nm × 80 μm included the diffraction signal from several grains at each mapped position and were collected on a Dectris Eiger X 4 M detector placed 104.9 mm downstream of the sample. Evaluation was performed using the pyFAI[78,79] software package and custom python scripts according to the methods described in refs. [80,81]. Compared to the referenced earlier works, the diffraction statistics were poor and therefore the evaluated strains and stresses represent a mixture of second-order (grain-averaged) values and first-order (polycrystal-averaged) values. In the case of the maps presented (Fig. 5) the values for the highlighted grain can be understood to represent second-order strain. The grain stands out from the surrounding material probably due to its comparatively large size.

### Electrochemical characterization

Electrochemical impedance spectroscopy measurements were performed with either a Novocontrol Alpha analyzer or a Solartron Modulab ECS XM with an FRA module exhibiting a range of 1 MHz. A 10 mV sinusoidal perturbation signal was applied in a frequency range of 1 MHz to 10 Hz with 10 points per decade after a 1 min open circuit voltage (OCV) period and the resulting signal response of the material was probed. A 10 mV signal was used due to the very small sample geometries present and the concern that galvanostatic electrochemical impedance spectroscopy (GEIS) may lead to Li deposition, especially at lower frequencies. For commercial batteries, which can have much lower impedance due to the large area, GEIS can result in a better signal to noise ratio[82]. Furthermore, it was found that for our specific setup GEIS is more vulnerable to interferences, especially at higher frequencies (see Supplementary Fig. 18). The combination of this led to the use of potentiostatic electrochemical impedance spectroscopy (PEIS).

Cycling under direct and pulsed current conditions (ms-, μs-regime) was conducted with either a Solartron ModuLab XM-unit or a 2450 SourceMeter from Keithley. For the determination of the applied current density, the apparent contact area was used (see Supplementary Note 10 and Supplementary Figs. 19 and 20). Li filament formation and propagation during electrochemical cycling was visually tracked with an optical microscope and images recorded on a 2–7 s period.

The electrochemical impedance spectroscopy measurements, plating proof of concept test and efficiency measurements (see Supplementary Note 6) of the PC-samples were conducted with a two-electrode Swagelok-setup, whereas plating experiments were conducted with a self-assembled coin cell setup as shown in Supplementary Fig. 21a, b. The estimated pressure in these setups is below 1.8 kPa. A homemade setup was also used for the electrochemical characterization of the SC- and HP-samples, where the pellet was clamped between two brass current collectors by means of a screw (see Supplementary Fig. 21c). All regular cycling experiments were conducted with a symmetric cell employing the Li:Sn alloy as both electrodes, whereas in case of the efficiency and plating experiments one side was switched for either Cu or Au, respectively. It should be noted that when contacting the Li:Sn electrodes only around $(1.8 \pm 0.8)$ kPa pressure was applied (details see Supplementary Note 2), which is three orders of magnitude lower than what is used during battery assembly[54,83,84] and that all measurements were performed in an Ar-filled glovebox under inert conditions ($O_2$ and $H_2O$ levels below <1 ppm) at $(21 \pm 1)$ °C.

### Modeling stress buildup inside a filament under pulsed plating conditions

In modeling the stress buildup within a metal filament, the authors used COMSOL Multiphyics 5.6. A Li metal electrode was modeled using elastic-rate-dependent plastic material behavior. The elastic and plastic material behavior were calibrated using data from Fincher et al.[85]. The elastic modulus was taken as 9.4 GPa, while the rate-dependent plastic behavior was modeled using a Norton Power Law Model, where the strain rate $\dot{\varepsilon}$ is related to the stress $\sigma$ through Eq. (4):

$$\dot{\varepsilon} = A \left( \frac{\sigma}{\sigma_{ref}} \right)^n. \tag{4}$$

Here, $\sigma_{ref}$ (a reference stress) is taken as 1 MPa, and fitting coefficients $A$ and $n$ were taken as 0.0052 and as 7.04, respectively. The sample was modeled as in the geometry shown in Supplementary Fig. 14, using red boundaries as "roller" boundary conditions. Under such a specification, the displacement normal to a "roller" boundary is set to zero. In this way, the resulting part effectively models a filament against an infinitely stiff solid electrolyte. Such a model would tend to experience larger stress buildups (and thus, potential relaxation) than one with a compliant solid electrolyte.

The temperature within the metal "filament" was prescribed as a function of time such that an equivalent amount of thermal strain was induced to the metal that would otherwise be plated. That is, one can define a "strain-rate" induced by plating as Eq. (5):

$$\dot{\varepsilon} = \frac{\dot{r}}{r_0} = \frac{V_m i}{F r_0}, \tag{5}$$

where $F$ is Faraday's constant, $V_m$ is the molar volume, and $r_0$ is the initial radius of the filament, $r$ is the radius of the filament and $i$ is the current density. The strain rate can then be defined as a function of time such that the temperature within the filament follows Eq. (6):

$$\dot{\varepsilon} = \frac{V_m i}{F r_0} = \alpha \dot{\Delta T}. \tag{6}$$

Where $\alpha$ is the coefficient of thermal expansion. Thus, by prescribing temperature as a function of time, the strain rate within the filament was controlled to match that of pulsed currents. The elastic energy in the filament was then calculated with Eq. (7) as:

$$\int \sigma : \varepsilon dV, \tag{7}$$

integrated over the body of the filament. The metal filament were meshed with free triangular mesh, for a total of 22,160 elements.

### Reporting summary

Further information on research design is available in the Nature Portfolio Reporting Summary linked to this article.

## Data availability

The data that support the findings of this study are available from the corresponding author on reasonable request.

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

## Acknowledgements

D.R. acknowledges financial support by the Austrian Federal Ministry for Digital and Economic Affairs, the National Foundation for Research, Technology and Development and the Christian Doppler Research Association (Christian Doppler Laboratory for Solid-State Batteries). Moreover, V.R. would like to thank the Austrian Marshall Plan Foundation for sponsoring parts of this project. P.F. and M.L.-M. acknowledge the European Research Council under the European Union's Horizon 2020 Program (FP/2014–2020)/ERC Grant Agreement (771834—POPCRYSTAL). The CSnanoXRD experiments were performed on beamline ID13 at the European Synchrotron Radiation Facility (ESRF), Grenoble, France. We are grateful to Manfred Burghammer at the ESRF for providing assistance in using beamline ID13 and to Pavan Badami at Argonne National Laboratory for providing calcined LLZTO powder for preliminary test measurements.

## Author contributions

D.R. designed the project. D.R., L.P., V.R. and F.F. performed the experimental work. S.G. synthesized the single crystals, Y.-M.C. and C.F. performed finite element calculations. S.W. performed the FIB SEM measurements. F.F. and J.T. performed the CS Nano XRD measurements with help from J.K. J.T. and J.K. analyzed the CS Nano XRD data M.L.-M. performed the AFM measurements with assistance from F.F. and the AFM data analysis with help from P.F. D.R. supervised the work. D.R., J.F., V.R. and F.F. wrote the first draft of the manuscript with discussion and feedback from I.H. and V.H. All authors contributed to the final draft.

## Funding

## Competing interests

The authors declare no competing interests.
