## [Peer Review File · Nature Communications]

REVIEWER COMMENTS

Reviewer #1 (Remarks to the Author):

The manuscript examines the impact of potential/current variation on electrochemically driven lithium metal growth in solid state electrolyte systems, in this study the LLZTO "garnet" system. The authors create a test platform to visualize lithium metal propagation through a solid state structure while applying the test waveform (figure 1). After various current interrogations the potential and impedance responses are correlated in time with visual evidence of lithium metal growth and asperity propagation (figure 2). The data is then assess and contextualized within the broader literature (figure 3).

Like all fracture studies in materials without dislocations, mechanisms are difficult to elucidate and cause, effect, and correlations can be deeply muddle. The care to preparation and efforts toward statistical significance of the manuscript are to commended, and the mechanistic hypotheses suggested by the pulsing duty cycle and frequencies bring some clarity to a method that is often a "black art."

This descriptor, in the main text, might be aided and distilled by a discussion of the "Sands Time", particularly with respect to figure 3d. Additionally, the critical current density discussion and mechanistic causality would be greatly benefited by a flow diagram: the cause and effect is in prose and the critical inflections listed in figure 3, but it takes significant effort even for the informed reader to thread the argument. A figure (4) flow diagram would aid greatly.

Reviewer #2 (Remarks to the Author):

The authors present a very intriguing method to mitigate Li dendrite formation. The method is novel and can be of importance for the practical application of Li metal anodes. The manuscript is well written and the methodology is sound in general.

However, there is one shortcoming that needs major revision before the manuscript can be published at all: Figure S7 shows more or less the basic problem of the study. The "end of plating" sample shows a much rougher surface than the "end of stripping" surface. Thus, the active surface area during the experiment must have changed. It does not matter if a DC or PP sequence is applied, but the constant current density labeled in S7 is simply not given. Holding 100 μm^2 of geometrical area is not the same as referring to the actual surface area for the contact of SE and Li metal.

One could even say that the authors compare apples to pears with the PP plating / DC stripping and PP plating and PP stripping. Since the applied current is constant for the pulse amplitude or DC part, the current density (μA per cm^2 of active area) is by far not the same as the actual current divided by the active surface area.

I suggest that the authors provide a significant amount of additional experiments evidencing the active surface area throughout the cycling (this could be done by chrono-amperometric techniques, substantiated by more ex-situ FIB-SEM images of the surfaces). Otherwise it is difficult to judge the full potential of the nonetheless intriguing pulsed current method.

Some other necessary improvements:

1) Using this many abbreviations as the authors did, does not help my understanding of the text at some places. I know that it can save some space, but for example in Figure 2 "... CCD values achieved by SC and HP samples during either DC or PP ..." is just tedious to go back and forth in the manuscript to recall what each and individual abbreviation stands for.

2) Besides, I did not find an explanation for what CCD stands for right away. Or at least I think that after searching for a few minutes without finding the meaning, it is not my task anymore but rather the authors duty to lay out to the reader more clearly.

3) Figure 1 is way too busy. Please have a look here <https://doi.org/10.1021/acscenergylett.2c01441> and present a less "noisy" graphic.

4) The authors are missing a large number of literature sources on the pulse current technique. (This approach is not new and has been applied to e.g. alkaline Zn metal anodes, non-aqueous Na/O₂ batteries and even gas diffusion electrodes.) Please give more insight for the reader than just reference [21].

5) Please improve the image quality of the SEM pictures in Figure S7.

Reviewer #3 (Remarks to the Author):

The authors demonstrated that the application of current pulses can improve critical current densities (CCD) compared to direct current (DC) processes. It was mainly found that pulse and pause effects reduce lithium concentration gradient in the solid electrolyte and avoid dendrite growth. The paper, as whole, is worthy of further significant refining for potential publication as a Nature Comm. However, similar type of approaches has been pursued in the literature and are worthy of a discussion in this study. The work that was published in ACS Energy Letters (ACS Energy Lett. 2020, 5, 3368–3373) where the authors showcased that the application of small current drives the healing of lithium dendrites in the solid electrolyte should be discussed. The one major missing confirmation in this study is truly the validation of the pulse concept in a full cell configuration, not only in symmetrical cells. Despite the promise, it appears that this work is incomplete. Specific comments for the authors are below:

1. Authors mentioned that the SSE is completely penetrated by lithium filaments and the voltage drops to around 0 V. In the subsequent half-cycle, they claim that the process is reversed as the filaments disappear and the sample appears to be intact. It should be noted that once the cell is electronically shorted, the lithium dendrites in the solid electrolyte cannot go back to the original state due to inhomogeneities in lengths, shapes, etc. The authors should explain this discrepancy and make sure that the observation is reproducible.

2. In Figure 9S, the authors show in the inset close-up view that the plating voltage comes down to -10 V while the stripping voltage went up but stayed less than + 5V. It appears that 5V is too much as a polarization voltage difference. So, either there is something wrong with the figure, or an error is associated with the experiment. The authors are invited to repeat the experiment.

3. The authors claim that the lithium filament can grow and pass through the single crystal (SC). The question is that how one can understand that lithium filaments can be initiated in

the single crystal from a mechanistic standpoint? Is it through the surface or through the interior of the SC? It is not clear from the provided images that the possibility of surface cracks can be ruled out. The authors should explain this by providing a deeper discussion.

4. In Figure 11S, the authors described that after the first short circuiting the lithium filaments form back, and the sample appears to be intact again. In the subsequent half-cycle, lithium filament grows once again through the upper area of the SC and short-circuits the cell. It is important to understand that lithium filaments do not grow all the sudden, but instead they grow gradually. So, under the same current the voltage polarization should not increase while filament growth is taking place, and one should even think that it should decrease. The opposite phenomenon is observed in the figure 11s. Furthermore, after shorting the cell the voltage is at 0 V and stays with time, meaning the stress relaxation of the filament should not reduce unlike what the authors claim. If there is misunderstanding, the authors should explain well these critical observations.

5. The authors should explain why they obtained very different effective CCD (150 and 450) in table 2S even though the cell is with the same ASR (10) and same pulse-pause ratio (1:3).

6. Since concentration/activity gradient is the main source of lithium filament growth and propagation, and since they also mentioned that pausing helps remove this gradient and increase the CCD, therefore, one can think that lower ratio of pulse-pause should give better results. The authors should explain why the opposite effect has been reported in the manuscript.

Response to Reviewer #1

Reviewer #1 (Remarks to the Author):

The manuscript examines the impact of potential/current variation on electrochemically driven lithium metal growth in solid state electrolyte systems, in this study the LLZTO “garnet” system. The authors create a test platform to visualize lithium metal propagation through a solid state structure while applying the test waveform (figure 1). After various current interrogations the potential and impedance responses are correlated in time with visual evidence of lithium metal growth and asperity propagation (figure 2). The data is then assessed and contextualized within the broader literature (figure 3).

Like all fracture studies in materials without dislocations, mechanisms are difficult to elucidate and cause, effect, and correlations can be deeply muddled. The care to preparation and efforts toward statistical significance of the manuscript are commended, and the mechanistic hypotheses suggested by the pulsing duty cycle and frequencies bring some clarity to a method that is often a “black art.”

-
- (1) This descriptor, in the main text, might be aided and distilled by a discussion of the “Sand’s Time”, particularly with respect to figure 3d.
-

Response: We really appreciate this comment, as it showed us that our argumentations connected to Figure 5b (prior: Fig. 3d) were slightly misleading the reader by sometimes referring to Li concentration instead of Li activity and therefore suggesting similarities to Sand’s time. We have adapted Figure 5b and the text accordingly and added a statement regarding Sand’s time to our discussion. As mentioned by Pang et al, (<https://doi.org/10.1016/j.joule.2017.11.009009>), the assumptions for Sand’s time are that there is a depletion layer of Li⁺ forming whereupon a local space charge creates a high electric field, leading to dendritic growth of Lithium. A single ion conductor, with a high transference number, preferably close to unity, should stop ion depletion and eliminate this electric field buildup. Cubic LLZO is such a single ion conductor and has a transference number close to unity, as shown by Buschmann et al (doi.org/10.1039/C1CP22108F). Therefore, Sand’s time is not applicable in our case and was not used for further explanation.

-
- (2) Additionally, the critical current density discussion and mechanistic causality would be greatly benefited by a flow diagram: the cause and effect is in prose and the critical inflections listed in figure 3, but it takes significant effort even for the informed reader to thread the argument. A figure (4) flow diagram would aid greatly.
-

Response: Thank you for the valuable suggestion. We agree with the reviewer's comment and designed an illustration (Fig. 4 in the main manuscript, see below) that allows the reader to follow the complex mechanism.

Fig 4. Proposed mechanism. Flow diagram showing the proposed mechanistic differences between direct current and pulsed current (μs) operation of a solid-state cell employing a Li metal anode: (1) pristine condition of the Li|SSE interface exhibiting a surface scratch of the SSE filled with Li, along with the lattice orientations of the adjacent grains near the defect tip. (2) Once the current is switched on Li-ions start to concentrate near the defect tip and (3) cause the buildup of an activation front, increasing over time t_n . LLZTO is thereby locally reduced and the lattice parameter in this region changed. (4a) In case of direct or long current pulse ($>\mu\text{s}$) application this continuous lattice distortion causes a continuously increasing amount of pressure which, at some point, is released in form of (5a) mechanical fracture. As a consequence, Li is plated along the new cracks (6a) and drives the mechanical fracture even further until a short circuit is caused. (4b) In case of short current pulses (μs) the time for the activation front buildup is short enough to just cause a minor lattice distortion before the current is switched off again. (5b) The accumulated Li-ions start to diffuse and distribute into the neighboring regions, hindering a significant pressure to arise. (6b) Afterwards, the current is switched on again like in (2) and the process repeats itself.

Response to Reviewer #2

Reviewer #2 (Remarks to the Author):

The authors present a very intriguing method to mitigate Li dendrite formation. The method is novel and can be of importance for the practical application of Li metal anodes. The manuscript is well written and the methodology is sound in general.

- **Major comments**

(1) However, there is one shortcoming that needs major revision before the manuscript can be published at all: Figure S7 shows more or less the basic problem of the study. The "end of plating" sample shows a much rougher surface than the "end of stripping" surface. Thus, the active surface area during the experiment must have changed. It does not matter if a DC or PP sequence is applied, but the constant current density labeled in S7 is simply not given. Holding 100 μm^2 per cm^2 of geometrical area is not the same as referring to the actual surface area for the contact of SE and Li metal.

Response: Thank you for your careful read of the manuscript. We have adjusted the associated text (Supplementary Note 6) and caption of Supplementary Figure 7 to clarify that for the plating experiments an asymmetric cell with a Li:Sn electrode and a Au electrode (imaged surface) was used to evaluate whether Li can be plated efficiently with 1 MHz pulsed currents. In the case of the cycling experiments, however, a symmetric cell with two Li:Sn electrodes was constructed, which is why the observed surface situation for this particular experiment does not reflect the situation present in the symmetric cells. Concerning the contact area issue, we agree with the reviewer that the geometrical area does not necessarily reflect the actual surface area quite correctly, as is also discussed by e.g. Fuchs et al (<https://doi.org/10.1002/aenm.202201125>), where the authors speculate that the contact area may be lower than the geometrically estimated one. Unfortunately, there are only few methods that can determine the actual contact area while being non-destructive, like e.g. nano X-Ray Tomography (<https://doi.org/10.1146/annurev-matsci-070616-123957> and <https://doi.org/10.1038/s41563-021-00967-8>). This technique, on the other hand, has a significant challenge in differentiating between Li (as it has a rather low attenuation coefficient) and void space like pores (see Shen et al. <https://doi.org/10.1021/acsenergylett.8b00249>), which was also an issue we ran into when we conducted similar experiments. Due to these still existing challenges, it is quite common in the field to determine the applied current density by the geometrical, apparent contact area, as is done e.g. by Krauskopf et al (<https://doi.org/10.1021/acscami.9b02537>). In this paper, they suggest the use of the apparent contact area which is why we have changed the wording in the main and SI document to apparent contact area.

(2) One could even say that the authors compare apples to pears with the PP plating / DC stripping and PP plating and PP stripping. Since the applied current is constant for the pulse amplitude or DC part, the current density (μA per cm^2 of active area) is by far not the same as the actual current divided by the active surface area. I suggest that the authors provide a significant amount of additional experiments evidencing the active surface area throughout the cycling (this could be done by chrono-amperometric techniques, substantiated by more ex-situ FIB-SEM images of the surfaces). Otherwise, it is difficult to judge the full potential of the nonetheless intriguing pulsed current method

Response: Thank you for your suggestions. Unfortunately, to perform chronoamperometric techniques one must precisely know the concentration of the mobile charge carrier, which, in comparison to a liquid electrolyte, is difficult to determine for a solid-state electrolyte as it is not necessarily equal to the Lithium concentration (<https://doi.org/10.1002/aenm.201500096>). The method to add a known quantity of a charge carrier as outlined by Emmel et al. (<https://doi.org/10.1021/acsaem.0c00075>) to use Chronoamperometry is not available for a solid electrolyte. One method to track the relative change of surface area is to use the change in interface capacitance, as explained by Krauskopf et al (see <https://doi.org/10.1002/aenm.201902568> and <https://doi.org/10.1021/acsaem.9b02537>). In the latter, it is mentioned that the load bearing contact area between LLZO and Li is not accessible. They assumed a completely covered initial surface, by applying significant pressure to Li (400 MPa) and used the negligible interface contribution as a confirmation of coverage. The same initial covered state applies to our samples as demonstrated by the Electrochemical Impedance Spectroscopy responses obtained for the SC Supplementary Fig. 5 and HP samples Supplementary Fig. 6).

Supplementary Fig. 1: (a) Impedance data of the SC samples measured prior to direct current (DC) or pulse plating (PP) experiments along with the respective (b) Bode plots.

To provide more insight into the interface condition of the cycled samples we have, however, performed additional SEM-FIB measurements as suggested by the reviewer (Supplementary Fig. 19 and below) and corresponding Electrochemical Impedance Spectroscopy measurements (Supplementary Fig. 20 and below). The images were taken of an HP sample cycled with either direct (a,b) or pulsed currents (c,d) at two locations to determine the potential changes at the Li|LLZTO interface. As evident from these images, no significant difference becomes evident between the two differently treated HP samples. In combination with the performed efficiency tests (Supplementary Fig. 8) we can therefore safely assume that the efficiency of Li⁺ transfer across the cell is not reduced by application of high frequency MHz currents and close to 100 %.

Supplementary Fig. 19: SEM images of FIB cuts performed at an HP|Li interface, at two different positions. In both cases, the Li electrode was coated afterwards with a protective Pt layer at the cut position for imaging purposes. Images were taken at 10 kV and 0.69 nA (a,b) or 0.17 nA (c,d) at a working distance of 4.1 mm. The HP sample treated with direct current is shown in subfigures a and b and was cycled until failure ($200 \mu\text{A}/\text{cm}^2$). Subfigures c and d show the interface of the HP sample cycled with pulsed currents until it reached the same current density as the sample treated with direct current. No significant difference becomes apparent as both interface regions show occluded voids and an intimate contact between LLZTO and Li. Therefore, it can be

assumed that the deposition of Li, as expected, works smoothly with pulsed currents.

Supplementary Fig. 20: (a) Nyquist plot showing the impedance evolution of a HP sample cycled with DC until failure. (b) Nyquist Plot showing the impedance evolution of a HP sample cycled with pulsed current application until the same current density as the sample in (a).²

- **Minor** comments.

- (1) Using this many abbreviations as the authors did, does not help my understanding of the text at some places. I know that it can save some space, but for example in Figure 2 "... CCD values achieved by SC and HP samples during either DC or PP ..." is just tedious to go back and forth in the manuscript to recall what each and individual abbreviation stands for.

Response: To improve the reading flow, the number of abbreviations has been reduced and kept to sample naming (HP, SC, PC) and terms which are common in the field and are used very often throughout the manuscript (CCD, LiBs, LLZO, RMS, ASR). Discarded: DC, PP, FIB-SEM, XRD, AFM, TEM; SEM, OM, SSLB

- (2) Besides, I did not find an explanation for what CCD stands for right away. Or at least I think that after searching for a few minutes without finding the meaning, it is not my task anymore but rather the authors duty to lay out to the reader more clearly.

Response: Thank you for mentioning that a vital part was not explained immediately, as it can be confusing and frustrating for the reader. An explanation for CCD has been added to the earliest mention in the introduction to remedy this. The part in question now reads. "Specifically, it is not clear to which extent and how pulsed current waveforms can increase the so-called Critical Current Density (CCD) of solid-state batteries, which is the current density up to which safe cycling can be conducted without the formation of Li filaments.^{7, 41, 42}

(3) Figure 1 is way too busy. Please have a look here <https://doi.org/10.1021/acsenergylett.2c01441> and present a less "noisy" graphic.

Response: Thank you for notifying us that Figure 1 did not paint a concise picture and confuses more than it helps. It has now been substantially reduced in complexity. The AFM pictures were split off and moved into the supplement as Supplementary Figure 3 and expanded to give more detail on their own. To further improve the image readability of the remaining figures in the main document, slight formatting changes were made as well.

(4) The authors are missing a large number of literature sources on the pulse current technique. (This approach is not new and has been applied to e.g. alkaline Zn metal anodes, non-aqueous Na/O₂ batteries and even gas diffusion electrodes.) Please give more insight for the reader than just reference [21].

Response: Thank you for notifying us on important literature on this topic. Additional sources in regard to the pulse current technique have been added.

- **more pulse plating**

21. Chandrasekar, M. S., and Pushpavanam M. Pulse and pulse reverse plating—Conceptual, advantages and applications. *Electrochim. Acta* **53.8** (2008): 3313-3322.
22. Hasannaemi, V. and Mukherjee S. highly catalytic amorphous Ni–P synthesized via pulsed electrodeposition. *Adv. Eng. Mater.* **21.7** (2019): 1801122.
23. Allahkaram, S. R., Golroh S., and Mohammadalipour M. Properties of Al₂O₃ nano-particle reinforced copper matrix composite coatings prepared by pulse and direct current electroplating. *Mater. & Des.* **32.8-9** (2011): 4478-4484.

- **Na-O₂ battery**

24. Langsdorf, D. et al. Pulse Discharging of Sodium-Oxygen Batteries to Enhance Cathode Utilization. *Energies* **13.21** (2020): 5650.
gas diffusion electrode
25. Konishi, N. et al. Electrochemical reduction of N₂O on gas-diffusion electrodes. *BCSJ* **69.8** (1996): 2159-2162.
26. Yang, H. et al. Effects of pulse plating on lithium electrodeposition, morphology and cycling efficiency. *J. Power Sources* **272** (2014): 900-908.

- **Zn metal anode and editor citation wishes.**

27. Grecia, G., Ventosa, E. and Schuhmann W. Complete prevention of dendrite formation in Zn metal anodes by means of pulsed charging protocols. *ACS Appl. Mater. Interfaces* **9.22** (2017): 18691-18698.
28. Xinrong H., et al. A review of pulsed current technique for lithium-ion batteries. *Energies* **13.10** (2020): 2458.
29. Qiao, Dongge, et al. Quantitative analysis of the inhibition effect of rising temperature and pulse charging on Lithium dendrite growth. *J. Energy Storage* **49** (2022): 104137.

(5) Please improve the image quality of the SEM pictures in Figure S7.

Response: We apologize for the confusion Supplementary Figure 7 has overall caused. As mentioned before the images were taken with an optical microscope, the missing scale bar has now been added to all images. Unfortunately, no vibrational table was available for the optical microscope situated in the glovebox, which is why due to vacuum pump vibrations no better image quality of the measurements can be provided. For visualization purposes, we have, however, increased the size of the images.

Response to Reviewer #3

Reviewer #3 (Remarks to the Author):

The authors demonstrated that the application of current pulses can improve critical current densities (CCD) compared to direct current (DC) processes. It was mainly found that pulse and pause effects reduce lithium concentration gradient in the solid electrolyte and avoid dendrite growth. The paper, as whole, is worthy of further significant refining for potential publication as a Nature Comm. However, similar type of approaches has been pursued in the literature and are worthy of a discussion in this study. The work that was published in ACS Energy Letters (ACS Energy Lett. 2020, 5, 3368–3373) where the authors showcased that the application of small current drives the healing of lithium dendrites in the solid electrolyte should be discussed. The one major missing confirmation in this study is truly the validation of the pulse concept in a full cell configuration, not only in symmetrical cells. Despite the promise, it appears that this work is incomplete. Specific comments for the authors are below:

Response: Thank you for your careful read of the manuscript. The work you mentioned (<https://doi.org/10.1021/acsenergylett.0c01896>) shows that it may be possible to remove dendrites grown at current densities exceeding the CCD by cycling for a longer time at very small currents. Whereas our approach aims to raise the CCD by varying the current application mode, tackling the root cause and not the symptom, in the form of dendrites. We would like to state, that it is also not known whether the healing of dendrites with low currents is viable in an asymmetric cell.

We agree with the reviewer that it would be of high interest to demonstrate the applicability of pulsed currents for electrochemical performance improvement in full cells. We are, unfortunately, up to this date restricted by significant limitations on the cathode side.

Current state of the art solid-state full cells using LLZO operate at C rates/current densities far below the current density levels investigated here. Specifically, these values currently lie at approx. 5 $\mu\text{A}/\text{cm}^2$ at RT or 100 $\mu\text{A}/\text{cm}^2$ at 100°C as shown in <https://doi.org/10.1016/j.joule.2018.02.007>, or 65 $\mu\text{A}/\text{cm}^2$ as shown in <https://doi.org/10.1149/1945-7111/ac644f>. Nonetheless, as a further validation of our method we have performed Nano XRD experiments done at the ESRF in Grenoble which show a locally generated strain at the dendrite tip, shown in Figure 5, which is also put below. This concurs with our hypothesis of an activity front changing the material stoichiometry near the defect and should theoretically also be valid for full cells. Proving this hypothesis would go beyond the scope of this paper but is a future project we would definitely like to pursue.

Fig. 5. **Strain at dendrite tip characterized by operando CSnanoXRD.** (a) The dendrite is highlighted in an optical microscope overview image, while the mapped region of $10 \times 10 \mu\text{m}$ is marked as a black-dashed square. The corresponding operando synchrotron X-ray nano diffraction strain maps show the region ahead of a Li dendrite directly after its growth and after a waiting time of approx. 10 min. A single grain is highlighted and exhibits a deviatoric strain of approx. 0.0045 (b), while it has relaxed to -0.0015 (c), equivalent to a change in deviatoric stress of approx. 750 MPa. This relaxation was not seen when comparing the state during the dendrite's growth and to the state directly thereafter. The grain is located approx. 8-10 μm in front of the assumed dendrite tip.

- (1) Authors mentioned that the SSE is completely penetrated by lithium filaments and the voltage drops to around 0 V. In the subsequent half-cycle, they claim that the process is reversed as the filaments disappear and the sample appears to be intact. It should be noted that once the cell is electronically shorted, the lithium dendrites in the solid electrolyte cannot go back to the original state due to inhomogeneities in lengths, shapes, etc. The authors should explain this discrepancy and make sure that the observation is reproducible.

Response: Thank you for bringing it to our attention when something is not clear enough. The case shown in the supplements in Figure S9 was included to demonstrate a specific peculiar behavior sometimes encountered during optical tracking of the SC samples. It appeared as if the formed dendrites, which shorted the sample, retract once the sign of the current is reversed in the following half-cycle and once again the cell generates a bulk response as shown in the Electrochemical Impedance Spectroscopy data in Supplementary Figure 13.

Another interesting aspect of this observation was, that in the subsequent half-cycle, Li dendrites would not necessarily regrow within these already created pathways but take a different route through the ceramic. Although this apparent “reversibility” of dendrites was a rarely encountered phenomenon, it was deemed important to show that such a circumstance is possible such that future observations of a similar behaviour have a frame of reference. The process itself is shown in more detail in the Supplementary video 3.

While it was not claimed that the sample is in its pristine state afterwards, we have updated the caption to avoid any further confusion and better reflect the uniqueness of this phenomenon.

(2) In Figure 9S, the authors show in the inset close-up view that the plating voltage comes down to -10 V while the stripping voltage went up but stayed less than + 5V. It appears that 5V is too much as a polarization voltage difference. So, either there is something wrong with the figure, or an error is associated with the experiment. The authors are invited to repeat the experiment.

Response: Thank you again for paying such close attention. As this is a symmetric cell both stripping and plating occur simultaneously. What happens in Supporting Figure 9 can be likely associated with contact loss during stripping, where the resistance of the cell and therefore the polarization increases due to the reduced contact area reaching close to -10V. This process is described rather well by Kasemchainan et al (<https://doi.org/10.1038/s41563-019-0438-9>), particularly in their Figure 4, which is also added below with the original caption.

Though they investigated a different solid electrolyte the principle should hold for the interface of LLZO as well. The polarization increase is also a sign of imminent failure and dendrite growth, incidentally used in the work you mentioned (<https://doi.org/10.1021/acsenergylett.0c01896>). In the subsequent cycles the contact area is increased again by lithium plating but also dendrite growth happening at the same time. The decreasing distance between the electrodes, caused by the growing dendrite, counteracts the lost surface area and the potential drops. Which is why the 10 V are not reached in the stripping voltage.

Fig. 4 (From Kasemchainan, J., et al. "Critical stripping current leads to dendrite formation on plating in lithium anode solid electrolyte cells." *Nature materials* 18.10 (2019): 1105-1111.) "Schematic of Li metal/Li₆PS₅Cl interface cycled at an overall current density above the CCS. Sequence from pristine to after several strippings and platings of Li metal: voids form on stripping; lateral growth of a thin Li film across the electrolyte surface (nucleating where void edges meet the electrolyte) on subsequent plating results in low plating polarization and formation of occluded voids. Some voids are eliminated on plating, but the occluded voids reopen on subsequent stripping leading to accumulation of voiding and increasing loss of contact with cycling. This increases stripping polarization and results in dendrites on plating."

- (3) The authors claim that the lithium filament can grow and pass through the single crystal (SC). The question is that how one can understand that lithium filaments can be initiated in the single crystal from a mechanistic standpoint? Is it through the surface or through the interior of the SC? It is not clear from the provided images that the possibility of surface cracks can be ruled out. The authors should explain this by providing a deeper discussion.

Response: Thank you for the comment. As shown in the SEM images in the now simplified Figure 3e, there is a subsurface crack in the single crystal, where the dendrite penetrated. The captions have been updated to clarify this matter. Moreover, it has been proven by some of the authors previously that dendrites do penetrate through the interior of the single crystal rather than at the surface (cf. Porz, L. et al. Mechanism of lithium metal penetration through inorganic solid electrolytes. *Adv. Energy Mater.* 7.20 (2017): 1701003).

- (4) In Supplementary Fig. 11, the authors described that after the first short circuiting the lithium filaments form back, and the sample appears to be intact again. In the subsequent half-cycle, lithium filament grows once again through the upper area of the SC and short-circuits the cell. It is important to understand that lithium filaments do not grow all the sudden, but instead they grow gradually. So, under the same current the voltage polarization should not increase while filament growth is taking place, and one should even think that it should decrease. The opposite phenomenon is observed in the figure 11s. Furthermore, after shorting the cell the voltage is at 0 V and stays with time, meaning the stress relaxation of the filament should not reduce unlike what the authors claim. If there is misunderstanding, the authors should explain well these critical observations.
-

Response: This matter is addressed in Supplementary video 4, as referred to in the supplements, which simultaneously connects the optical observations to the respective polarization. It is shown that the polarization does not increase while filament growth takes place and can likely be attributed to contact loss on one side. In the subsequent cycle filament growth starts and the polarization drops as a result. The captions have been updated to better reflect this and an additional mention to supplementary video 4 has been added. The video also shows that once the cell is shorted, the ceramic is cracked and the polarization drops to 0 V. Therefore, stress relaxation as mitigation for filament propagation and fracture of the solid electrolyte is not relevant anymore.

- (5) The authors should explain why they obtained very different effective CCD (150 and 450) in table 2S even though the cell is with the same ASR (10) and same pulse-pause ratio (1:3).
-

Response: Supplementary Note 3 in the SI was clarified to better reflect the state of the samples discussed. Overall, the samples in question were poly-crystalline LLZTO samples prepared via solid state synthesis for initial test measurements and did not undergo an extensive surface polishing procedure (as opposed to the poly-crystalline LLZTO samples used for the main measurements). Since the ASR does not reflect the presence of small surface cracks, which when filled with lithium are a prime spot for current focusing and filament growth, varying sample quality is the most likely explanation for the varying electrochemical performance. We have added an explanation regarding this observation to Supplementary Note 3.

- (6) Since concentration/activity gradient is the main source of lithium filament growth and propagation, and since they also mentioned that pausing helps remove this gradient and increase the CCD, therefore, one can think that lower ratio of pulse-pause should give better results. The authors should explain why the opposite effect has been reported in the manuscript.
-

Response: This is tied to comment 5, which is now further elaborated in supplementary note 3. The samples where the different pulse:pause durations were tested were of lower surface quality than the ones used for the main experiments. The surface quality can overshadow the effects of an applied current program, be it direct or pulsed current, which is why a large focus was placed on keeping the surface quality as high as possible in the main experiments. Even with this there is still a large sample variation visible. The reviewer is right that the lower ratio of pulse:pause experimentally showed slightly better CCD results. However, in order for longer pause times to remain competitive with direct current or smaller pulse times, the applicable current density has to increase with the same ratio. e.g. a 1:10 pulse pause ratio needs a 10 times higher CCD than direct current application otherwise overall charging is slower and not viable. Since this tendency was not visible in the samples with a lower surface quality as a matter of time the higher pause durations were not further investigated.

REVIEWERS' COMMENTS

Reviewer #1 (Remarks to the Author):

All of my questions are addressed satisfactorily.

Reviewer #2 (Remarks to the Author):

The authors have replied to the major shortcoming comprehensively. They provide additional experiments (EIS and FIB-SEM) to prove that the apparent contact area seems to be comparable for the different samples and techniques. The changed wording in the SI and the main part of the manuscript now helps the understanding of this fine detail. All minor recommendations have been implemented satisfactorily as well. Figure 1 is much much clearer now. (Besides, adding Figure 4 with the proposed mechanism now really helps the reader to understand the proposed strategy in the manuscript.)

I now can recommend the manuscript to be accepted for publication in Nat. Comm.

Reviewer #3 (Remarks to the Author):

The authors have done a decent job in addressing the comments of the reviewers. The quality and impact of the paper raised significantly to warrant a publication in nature communication.